# Nitric Oxide in the Field: Prevalence and Use of Nitrates by Dietitians and Nutritionists in Spanish Elite Soccer

**DOI:** 10.3390/nu15245128

**Published:** 2023-12-16

**Authors:** Jaime Sebastiá-Rico, Daniel Cabeza-Melendre, Liam Anderson, José Miguel Martínez-Sanz

**Affiliations:** 1Area of Nutrition, University Clinic of Nutrition, Physical Activity and Physiotherapy (CUNAFF), Lluís Alcanyís Foundation—University of Valencia, 46020 Valencia, Spain; 2Food & Health Lab, Institute of Materials Science, University of Valencia, 46980 Paterna, Spain; 3School of Sport, Exercise and Rehabilitation Sciences, University of Birmingham, Birmingham B15 2TT, UK; l.anderson.4@bham.ac.uk; 4Nursing Department, Faculty of Health Sciences, University of Alicante, 03690 Alicante, Spain; josemiguel.ms@ua.es

**Keywords:** ergogenic aids, soccer, football, nitrates, nitric oxide, beetroot juice

## Abstract

Soccer players make frequent use of dietary supplements to improve performance. One of the most widely used strategies to optimize performance is to increase the bioavailability of nitric oxide through nitrates, as it could delay fatigue during physical exertion, among other benefits. This may be positive for performance in soccer, although there is almost no research in professional soccer. The aim of the study was to evaluate the use of nitrates and behaviours related to their consumption in Spanish elite soccer clubs. Dietitian–nutritionist representatives from 45 teams from the most important Spanish soccer leagues completed an online survey to determine if, when, how and why nitrates are prescribed to soccer players. Of the total sample, 55.6% indicated providing nitrates, always before matches, but only 36% in training. There was a wide variation and lack of consistency in the timing, dosage and form of administration of nitrates. The use of mouthwashes or the protocol of chronic nitrate intake was not taken into account in most cases. The present study indicates a lack of interpretation between scientific knowledge and its application in practice, highlighting the need for future research to better understand how to optimize the use of nitrates in professional soccer.

## 1. Introduction

Soccer is a team sport played by men and women that is in constant evolution, as it has experienced an increase in the physical and technical demands, as well as increased economic implications of winning or losing [1]. As a result, training regimes have become more demanding and sophisticated, to prepare players to meet the demands of the games [2]. The high weekly frequency of matches that increase the risk of injury, the wide variability of competition schedules for television purposes, the cultural diversity within each team and the amount of travel to play matches in different competitions, both national and international, are some of the different difficulties that today’s elite player has to overcome [3,4,5]. All these factors contribute to a change in the dietary needs of soccer players, and they must be addressed by dietitians or nutritionists [5,6].

Players during training and matches work at low and high intensities, including intermittent exercises of prolonged duration such as walking, jogging, running at high and low speeds, sprinting, moving in reverse, kicking, jumping or tackling [7]. Encompassing all of these actions, elite players also run a total distance of between 9 and 14 km during match play [5,8]. It is estimated that around 70% or more of the actions performed during a match are of low intensity [7], while covering 22–24% of the total match distance at speeds above 15 km/h, 8–9% at more than 20 km/h, and 2–3% at more than 25 km/h [9]. Furthermore, it has been calculated that the average oxygen uptake during a match is around 70% of the maximum oxygen uptake, while the heart rate is maintained at an average of 85% of the maximum [5,7].

Supplementation can be very useful for general health promotion, injury prevention and performance enhancement, especially in sporting contexts, where a small improvement can make a big difference [5,10]. Although supplements are widely known and consumed by most athletes, not all supplements have the benefits advertised or desired by athletes [10,11]. The Australian Institute of Sport (AIS), a reputable entity within the scientific community, elucidated the ABCD classification of the most commonly used supplements and ingredients in sports practice, dividing them into four groups based on the scientific evidence shown and respecting the efficacy, legality and safety of their application [12]. Among the supplements that fall into category A are nitrates. Nitrates are a promising supplement but are underused and little studied, compared to other supplements both in the general population and in elite athletes, either due to lack of knowledge or because of the protocol for their use [5,12,13].

Nitrates are found in many foods, the major source for human consumption being vegetables such as spinach, beets and arugula [14]. In recent years, there has been growing interest in the possible effects on performance enhancement and post-exercise recovery following nitrate consumption, as it may improve blood flow during physical exertion, reducing oxygen demand and improving mitochondrial function, among other physiological impacts [15]. Currently, there is little information about nitrate intake patterns among high-level soccer players [16,17,18]. This lack of knowledge makes it difficult to determine whether good practices are being followed in terms of nitrate use and safety assessment, which can lead to following excessive and/or inappropriate nitrate dosing strategies, as well as choosing low-quality products whose nitrate content falls below the minimum ergogenic dose (≤5 mmol·d^−1^) to obtain the expected physiological benefits, leading to unnecessary financial expense or even side effects such as drug interactions or digestive symptoms [12,13,19].

With this in mind, the main objective of this research was to examine the patterns of dietitian and/or nutritionist recommendations and behaviours associated with nitrate consumption in Spanish elite soccer teams. It was initially hypothesized that there will be differences in the dose, form of administration and timing of prescribed nitrate consumption and practitioners will not follow supplementation guidelines for nitrate consumption for soccer.

## 2. Materials and Methods

### 2.1. Study Design

This is a descriptive and cross-sectional observational study on the patterns and protocols for the dietary prescription of nitrate supplementation in Spanish elite soccer teams. The research population was chosen by means of a non-probabilistic convenience sampling among the Spanish football teams that had a dietitian or nutritionist as part of their medical staff. The study was approved by the Ethics Committee of the University of Valencia (code: 1534145), in accordance with the World Medical Association codes and the Declaration of Helsinki.

### 2.2. Participants

All dietitians (dietetic technician, 2 years of academic training in Spain) and nutritionists (university degree, 4 years of academic training in Spain) incorporated in the squads of Spanish soccer teams, both professional and semi-professional, were contacted electronically on three occasions to participate in the study (N = 105). Data were obtained from a final sample of 45 Spanish elite soccer teams (Liga Santander; N = 8, Liga Smartbank; N = 10, 1st Real Federación Española de Fútbol (RFEF); N = 11, 2nd RFEF; N = 11, 3rd RFEF; N = 5, 1st RFEF Femenina; N = 8, 2nd RFEF Femenina; N = 3). The survey was answered by the club nutritionists (N = 40) and dietitians (N = 5). Participants were asked to give informed consent before agreeing to and completing the survey questions.

### 2.3. Survey Design

The survey required the participants to describe typical club nitrate supplementation practices. A self-administered online survey was developed for obtaining the results (Appendix A: Use of nitrates as an ergogenic aid in elite football). This survey was prepared, agreed upon, tested and administered by the research team using a Google Form (https://forms.gle/18tCu1bJvrfdxYnj9 (accessed on 14 December 2023)) and sought to determine if, when, how and why nitrates were prescribed to players using different multiple choice, pick list, grid and free text questions, based on a similar study that evaluated the use of caffeine [20]. Two separate survey tracks were designed, to distinguish clubs that prescribe nitrates and those that did not. Responses with reported adverse effects of nitrates and the perception of nitrates as an ergogenic aid were based on previous evidence indicating that nitrates may elicit a response in each parameter included. The final survey was available in Spanish and was sent out in March 2023. Those who chose to participate accessed the survey through a secure link sent in the recruitment email.

### 2.4. Statistical Analysis

Descriptive data were reported and presented as response frequency. Key themes were distinguished from free text questions, and the frequency of each theme was shown in the results.

## 3. Results

Of the clubs sampled, 56% (N = 25) provide nitrates to players, with almost half of the teams (N = 12) having been administering nitrates to their players for a period of 1 to 3 years (Figure 1A). The primary decision on their recommendation is based on published scientific literature (N = 23; 92%), followed by personal experiences with other athletes (N = 11; 44%) and the recommendation of fellow professionals (N = 6; 24%). The data indicate that all teams who provide nitrates to players, administer before the match, with the most frequented time of consumption being 2–3 h before the competition (N = 17; 68%), followed by 1–2 h before and 3 or more hours before matches, which share the same frequency (N = 4; 16%); only two clubs follow the strategy of administering nitrates to their players 1 h or less before the match (N = 2; 8%) (Figure 1B).

However, in training, nitrate supplementation is less frequent (N = 9; 36%). In addition, some of the teams have used them in training to test the supplement sporadically (N = 6; 24%). The most common time for the prescription of nitrates in training is around 1 h or less before (N = 4; 16%), with variability in the data, as some clubs administer nitrates 1–2 h before, 2–3 h before or even 3 or more hours before training, sharing the same frequency (N = 1; 4%). There is a club that administer nitrates to their players after training in the next hour or 2 h after training (N = 1; 4%) (Figure 1C). In addition, 72% of the clubs (N = 18) do not perform nitrate loading with players, but the most followed strategy among those that do is nitrate loading for 3 days prior to the match (N = 3; 12%) (Figure 1D).

The forms of nitrate administration are diverse, with shots (N = 17; 68%) and powdered nitrates (N = 16; 4%) being the most popular, followed by supplementation via capsules (N = 8; 32%) (Figure 2B). Seventeen teams (68%) indicated that they use more than one form of nitrate administration, and eight (32%) use only one form (Figure 2A). The most frequently used supplementation brand to deliver nitrates was Scientific Nutrition (N = 18; 72%), followed by Fullgas (N = 5; 20%) (Figure 2C).

More than half of the teams (N = 13; 52%) do not monitor nitrate intake through the athletes’ regular diet. Twenty-two clubs (88%) stated that they prescribe the same amounts of nitrates with their players, although most teams do not all follow the same consumption strategy (N = 14; 56%). The following are open-ended responses to the reasons why not all soccer players have the same consumption pattern (Table 1):

Of the clubs that administer nitrates to their players, twenty (80%) provide absolute doses, three of them target a range of doses (12%) and two (8%) teams report not having a clear dose, either due to lack of knowledge, or to be determined according to the context. The absolute dose range provided is between 250 mg and 560 mg of nitrates, with the most frequently reported amount being 500 mg (N = 13; 52%), followed by 360 mg (N = 3; 12%) and 400 mg (N = 2; 8%). There is a difference between the three clubs reporting unspecified doses, whose ranges range from 300 to 600 mg, from 350 to 500 mg and from 450 to 600 mg, respectively (Figure 3A). In addition, 88% (N = 22) of the sampled teams using nitrates provide them together with other supplements, specifically with up to 17 different ones. The supplements most commonly used in combination were caffeine (N = 16; 64%), creatine, gels and isotonic drinks (N = 12; 48%) (Figure 3B).

Seventeen clubs (68%) do not control the use of mouthwashes or mouth rinses. However, 28% (N = 7) do control them around the use of nitrate supplementation and only one team (4%) takes them into account on a daily basis. Adverse effects, due to nitrate consumption, were observed in twenty-two clubs (88%), the most frequent being urine colour change (N = 18; 72%), bad taste (N = 14; 56%) or gastrointestinal problems (N = 8; 32%) (Figure 4A). In contrast, reported side effects occurred occasionally or extremely rarely (N = 21; 84%) (Figure 4B).

Improved endurance performance or delayed fatigue during physical exertion (N = 22; 88%) and improved muscle strength (N = 9; 36%) were the ergogenic effects most needed to be obtained with nitrate intake (Figure 5).

## 4. Discussion

The main objective of this study is to present a first analysis of the frequencies, practices and perceptions of the use of nitrates as an ergogenic supplement in Spanish professional soccer. The main findings of this study were: (1) the prescription of nitrates as an ergogenic aid is frequent in competitions in the main Spanish professional soccer leagues but not so common in training; (2) most clubs do not take into account mouthwash control, nor do they monitor nitrates through diet; (3) nitrate loading strategies are not commonly used; (4) various forms of administration, times of consumption and doses are used.

### 4.1. Nitrate Consumption in Elite Soccer

The data from this study indicate variety in the mode of administration of nitrates, with vials or shots of beetroot juice (BRJ) being the form of prescription most used by soccer teams, followed by powdered nitrates (Figure 2B). This result, despite the limited number of studies evaluating the efficacy of different forms of nitrate administration such as nitrate salts, may be consistent with the scientific literature because they allow direct contact with nitrate-reducing bacteria, thereby increasing the bioavailability of nitric oxide (NO) [21,22,23]. However, the same is not true for the capsules, since, in the process of digestion, they could avoid direct contact, where part of the conversion of nitrates to nitrites occurs [15]. Finally, homemade BRJ do not seem to be the optimal choice, as the nitrate content can vary considerably depending on growing conditions and time since harvesting [24], and it has been found that their nitrate content can decrease depending on whether they come from organic crops and/or if they have been cooked previously [25].

At the same time, there is controversy about the raw material used to obtain the dose of nitrates. In order to improve the adherence of athletes, several professionals prescribe amaranth, reducing the pre-exercise consumption time from 2 h to 1 h, but the plasma concentration of nitrites seems to be irregular, compared to the pharmacokinetics of concentrated BRJ, reducing the net amount destined to be converted into NO [26]. The amaranth is rich in potassium and an antioxidant called amaranthine and does not contain oxalates, whereas beet does [27]. However, according to the current literature available, amaranth does not appear to be the best dietary source of nitrates, but studies are needed where doses are administered in the ergogenic range [27,28,29,30]. In contrast, beet is rich in vitamin C and phenolic compounds (especially flavonoids), which are a great help to avoid oxidative stress and protect NO, increasing its biological actions in the organism [19,31]. Recently, beet consumption has been found to trivially improve endurance exercise performance; no significant effects were observed after consumption of other nitrate-rich vegetables such as amaranth, chard or rhubarb [32]. Table 2 presents a compilation of products rich in nitrates from the brands that were mentioned by nutritionists and dieticians in the questionnaire. The table highlights the amount of nitrate per service of product. In conclusion, the authors consider that could be wrong to extrapolate ergogenic results from BRJ to amaranth, because both foods contain nitrates without considering the food matrix.

The results seem to indicate that the use of nitrates is common in Spanish professional soccer leagues, based on scientific research as the main reason for use among those responsible for prescribing nutritional supplements to players. However, although there is unanimity in their use in matches, only nine teams that do prescribe nitrates provide them in training sessions (Figure 1C). The great variability in the results referring to the moments of consumption, as well as the dose used and the form of administration of nitrates, seems to indicate that the strategy of use is based on the individual self-prescription preferred by the athlete [13]. These different approaches to nitrate intake may be due to the fact that there are currently few studies that determine the effects of nitrates in soccer, resulting in limited knowledge about optimal nitrate intake strategies. However, there are several studies on the use of nitrates in soccer players that suggest benefits, such as improvements in power, muscle function and performance in physical tests [5,16,17,33,34]. Further research is needed to fully understand the effects of nitrate consumption in professional soccer, as positive results in sports performances following nitrate consumption have not been consistently observed [13,35]. Additionally, when utilizing dietary inorganic nitrate as an ergogenic aid, the level of aerobic fitness must be taken into account. Research has demonstrated that individuals with a maximal oxygen uptake (VO2peak) of less than 45 mL/kg/min exhibit a greater ergogenic effect with an acute nitrate intake, while chronic nitrate supplementation is also ergogenic for individuals with a VO2peak below 60 mL/kg/min, including those with low to moderate aerobic fitness levels [13,23,35,36,37]. Therefore, it is advisable to consider the individual’s level of aerobic fitness when contemplating the use of dietary inorganic nitrate as an ergogenic aid, particularly since professional soccer players tend to be individuals with high aerobic fitness.

The data obtained regarding the time of consumption appear to be very broad, ranging from less than one hour prior to the match to more than three hours prior to the match, with consumption 2–3 h prior to the competition being the most frequent (Figure 1B). The maximum concentration of nitrites and nitrates in plasma occurs approximately 1 and 2.5 h, respectively, after acute ingestion of nitrates, and they remain for approximately 5 to 6 h in circulation [13,38]. There is a clear dose–response relationship between the amount of nitrate consumed and the magnitude of subsequent peak plasma nitrite and nitrate concentrations [13,39]. Considering the administration times described in the present study, they seem to be appropriate to reach the maximum concentration in blood plasma and thus take advantage of the expected ergogenic effects [13,38].

In addition, elite athletes often consume nitrates in combination with other ergogenic aids. The most commonly used ergogenic aids in our study were caffeine, isotonic drinks, gels, creatine and electrolytes. The literature on the ergogenic effects of nitrate co-administration with other ergogenic aids is limited, but suggests that it is important to take into account the type and intensity of exercise, as well as chronic nitrate intake, to optimise the benefits of supplementation [40].

On the one hand, only eight clubs control the use of mouthwashes in their players, compared to seventeen that do not. Taking into account the importance of the nitrate-reducing microbiota in the oral cavity for the final conversion to NO, it has been shown that the use of mouthwashes reduces nitrite concentrations in both saliva and plasma, thus decreasing the reduction capacity of oral nitrate [23,38,41,42]. This results in a lower bioavailability of NO and, as a consequence, a decrease in the desired ergogenic effects [43]. In addition, factors such as gender, age, ethnicity, papillary structure of the human tongue, dietary intake of nitrates, tobacco, oral hygiene habits, oral temperature, pH or certain pathologies seem to condition the oral microbiome [15,23,44]. It has also been observed in a study with elite athletes that a low-carbohydrate diet pattern resulted in a significant reduction in the abundance of bacteria of the *Haemophilus*, *Neisseria* and *Prevotella* families, coinciding with a loss of exercise economy by increasing oxygen cost during exercise, compared to a moderate, high-carbohydrate diet [45]. It is essential to consider all these factors, as they are determinants in making the best decisions regarding nitrate prescribing.

In relation to dosage, there were only three buckets (12%) that administered unspecified doses in three different ranges between 300 and 600 mg. Players from 18 teams (52%) consumed absolute doses of 360 mg, 400 mg and 500 mg (range 250–560 mg). It is widely accepted that any dose between 5.1 and 9 mmol/d^−1^ could elicit an ergogenic effect, although it has been shown that taking more than 10–12 mmol/d^−1^ is no more effective [36]. The majority of clubs using nitrates (52%) do not monitor their intake from food, and 72% of teams do not perform nitrate loading. Recently, it has been discovered that skeletal muscle can act as a reservoir for nitrates and nitrites [46]. This means that the muscle is able to store them from the diet and could influence the efficacy of their supplementation. In addition, the presences of sialin, an active nitrate transporter, and xanthine oxidoreductase have been found in human skeletal muscle [47]. This suggests that muscle has the necessary mechanisms to store, metabolize and transport nitrate from the diet, contributing to NO generation during exercise. Because of this, it is important to control the nitrate content in the athlete’s habitual diet, as well as to perform nitrate loading protocols [13].

Several studies have examined the impact of nitrate intake on sports performance for both men and women, exploring any potential differences based on gender [48,49]. There is evidence supporting its effectiveness in improving strength and biomechanical functions in both genders [48]. Improvements in exercise economy and endurance capacity were also observed after consuming BRJ among healthy young adult males; however, no such benefits were seen in females [49]. Nitrates have also been studied, regarding their use in female athletes [50,51,52,53,54]. No ergogenic effects were observed on oxygen consumption, heart rate or aerobic time trial performance after the consumption of beetroot juice in female athletes [52,53]. Additionally, in a single study, BRJ intake not only failed to enhance exercise economy and vascular health but also had an adverse effect on aerobic endurance capacity [54]. However, an increase in ratings of perceived exertion and increased plantar flexor torque production were observed [52,53].

In female soccer, BRJ consumption for 6 weeks was positively associated with improvement in haematological parameters related to red blood cell count and iron [50], but in another study where the same dose of BRJ was used but acutely, it did not show beneficial results in these haematological parameters [51]. In summary, the potential effect in women remains to be clearly established. Nonetheless, the absence of favourable outcomes among women could be attributed to methodological considerations and not necessarily to sex differences, as future studies should take into account individual factors such as menstrual cycle, oral contraceptive use, oral hygiene, and nitrate dosage, as well as include groups of women with different aerobic capacities to obtain clearer results on efficacy [55]. Finally, this study’s results have led to the authors including recommendations on the utilization of nitrates in soccer, based on referenced scientific literature (Figure 6).

### 4.2. Adverse Effects of Nitrate Consumption

No frequent side effects have been described, and the most predominant adverse effects were minor, such as pink urine colour (18; 72%), an unpleasant taste (14; 56%) or gastrointestinal discomfort (8; 32%) (Figure 4A). However, the latter two may be important, as they may limit its use in practice by generating rejection from players. Both adverse effects seem to depend on the subjective characteristics of each individual, as well as the physiology of each player [12,13]. Finally, if a soccer player is not accustomed to taking nitrates and only takes them on game day, they may be more likely to cause gastrointestinal symptoms. As with other supplements, it is recommended that athletes become used to taking them occasionally during training.

### 4.3. Perception of Nitrates as an Ergogenic Aid

The main reason for the use of nitrates as an ergogenic aid in Spanish soccer was the delay of fatigue (88%), followed by the improvement in muscle strength (36%) (Figure 5). NO obtained through the nitrate–nitrite–NO pathway acts as a vasodilator, improving mitochondrial efficiency and reducing oxygen cost during exercise [56]. In addition, this reduction occurs to a greater extent under certain conditions of acidity and hypoxia [57], so that an increase in blood lactate may increase NO bioavailability. Because of this, the consumption of nitrates benefits athletes of high-intensity intermittent sports, such as soccer, to a greater extent. However, there are clubs that prescribe nitrates in order to improve the health of their players, as their consumption has been shown to be beneficial for vascular health [13,58,59].

### 4.4. Limitations

The present study provides valuable data on the prevalence and practices of nitrate consumption in professional soccer, but there are certain limitations. First, it was not possible to obtain data from individual players in this study. In addition, not all clubs have dietitians or nutritionists, nor did all teams with these professionals respond to the survey, potentially due to its nature of answers for convenience. Despite knowing the brands of supplementation, it was not possible to know the specific product administered in the teams, in order to know the nutritional information of that supplement. Another limitation of the study is the lack of validation of the participant survey, despite the fact that it was developed, agreed upon and tested previously by the research team based on a similar survey and population on caffeine supplementation [20]. Finally, a specific dose was also not described in many teams, as they provided ranges of nitrate doses and there may be a large difference in the dose administered.

## 5. Conclusions

Nitrate supplementation is common in the sample of dietitians and nutritionists from the main Spanish leagues, but its prescription is variable. Their prescription is mainly limited by club decisions, and those clubs that use them seem to follow a consumption strategy focused only on matches, leaving aside their use in training. In addition, strategies focused on adequate nitrate utilization such as multi-day loading, and practices such as the use of oral mouthwashes or monitoring of nitrates through food, are not usually taken into account. Finally, there is great variety in the timing, dosage and form of supplementation administration, so it is recommended that clubs follow the protocol of use indicated by the AIS or by expert groups.

## Figures and Tables

**Figure 1 nutrients-15-05128-f001:**
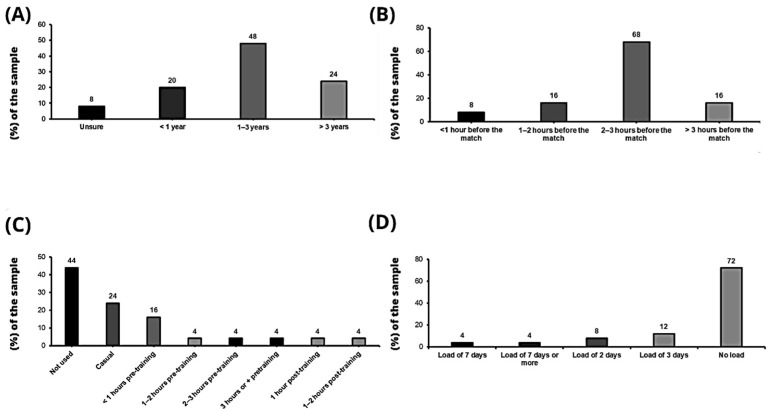
(**A**) Duration in which nitrates have been used as a supplement; (**B**) when nitrates are prescribed in matches; (**C**) when nitrates are prescribed in training; (**D**) nitrate loads.

**Figure 2 nutrients-15-05128-f002:**
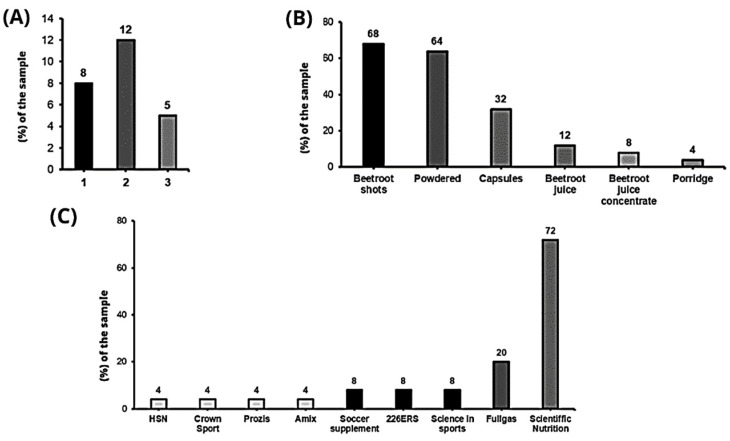
(**A**) Number of nitrate consumption methods; (**B**) nitrate consumption methods used; (**C**) products by commercial brands used.

**Figure 3 nutrients-15-05128-f003:**
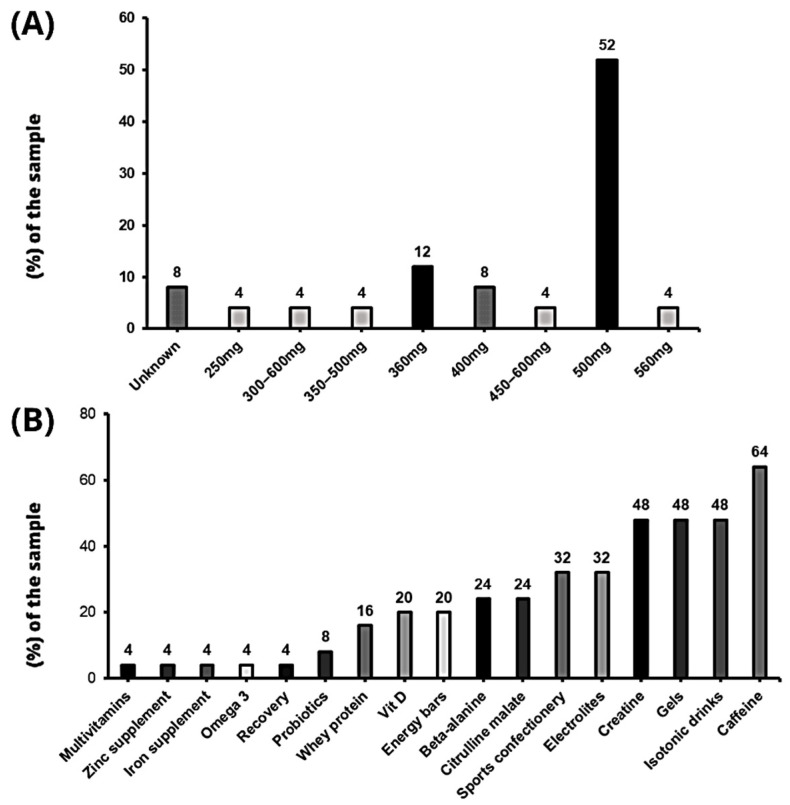
(**A**) Administered dose of nitrates; (**B**) Other supplements consumed along with nitrates.

**Figure 4 nutrients-15-05128-f004:**
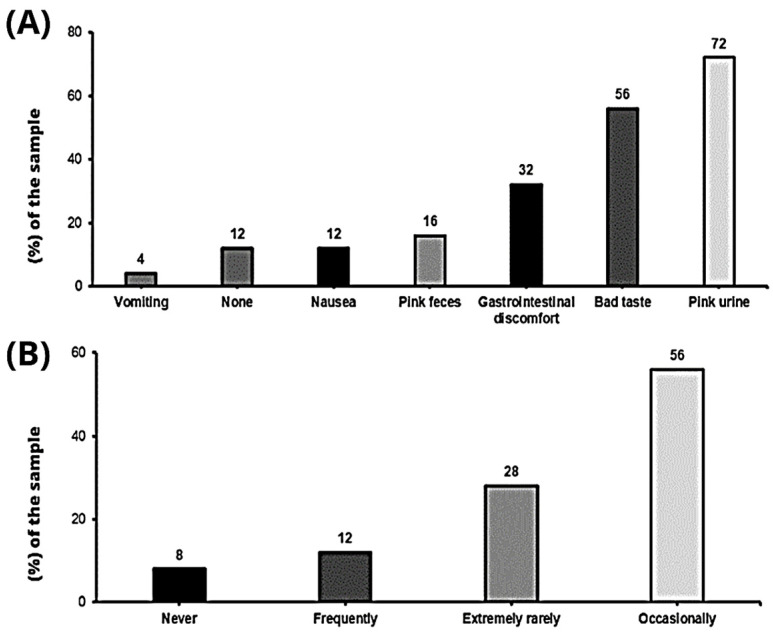
(**A**) Adverse effects reported after nitrate consumption; (**B**) frequency of adverse effects reported.

**Figure 5 nutrients-15-05128-f005:**
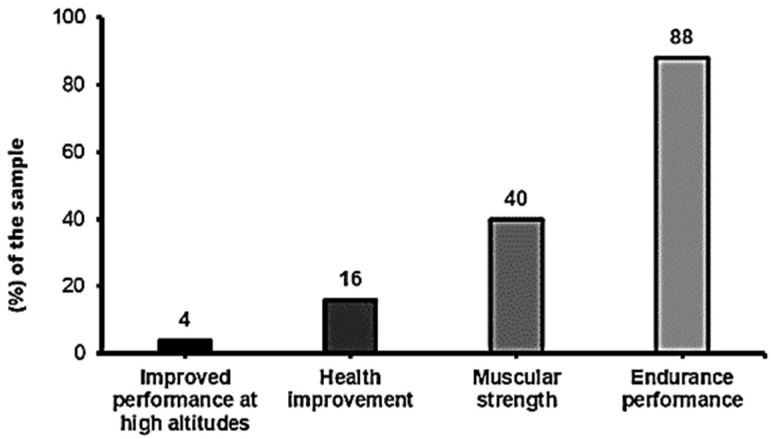
Purposes of nitrate consumption.

**Figure 6 nutrients-15-05128-f006:**
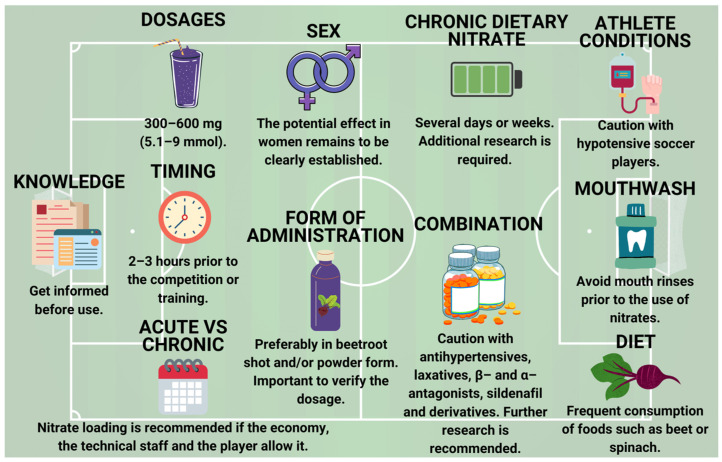
The 11 headlines on the use of nitrates in soccer.

**Table 1 nutrients-15-05128-t001:** Compilation of responses regarding nitrate consumption strategies.

They are used to recover from injuries, not only in matches or in periods of many matches.
Some take it only in the pre-matches. Others in matches and in training.
Some do not consume them because of unpleasant taste or side effects.
Some in shots. Players who find the taste of the shots unpleasant consume the shot in the form of a banana shake with powdered nitrates and water. Some with gelatin and others in the form of capsules.
Only players called up on the day of the match are used.
It is the players who pour the glass with the nitrates and not all of them take them. Therefore, they are not always the same doses.
They are only used by 2–3 players on a continuous basis.
Some do not take them because of gastrointestinal discomfort, others because of unpleasant taste, others do not remember to consume them or do not want to do so.
They are used together with caffeine and it is difficult to establish an individual dose.
In more demanding matches, depending on the position: Goalkeepers take it pre-match. Forwards and midfielders perform nitrate loading during training and matches.
Some, apart from the match, usually perform loads 1 or 2 days before the match.
Players are recommended to take 400 mg of nitrate powder 1 h before the match.
They try to adjust dosage, administration time and presentation to what the players will accept.
Same dose and moment of consumption, but mix with other beverages according to personal taste.

**Table 2 nutrients-15-05128-t002:** Compilation of nitrate-rich products from the brands mentioned in the survey.

Brand	Product Name	Nitrates per Service
Scientific Nutrition	Nitrates shot	500 mg per 60 mL (one shot)
Scientific Nutrition	Nitrates +	459 mg per 33 g
Fullgas	NOS Nitrate	360 mg per 18.5 g
Science in Sport	Performance Nitrate Shot	500 mg per 60 mL (one shot)
Science in Sport	Performance Nitrate Powder	500 mg per 55 g
Science in Sport	Performance Nitrate Gel	250 mg per gel
226ers	Nitro Pro Beetroot	400 mg per 10.3 g
Soccer Supplement	Pro Grade Nitrate Shot	400 mg per 60 mL (one shot)
Amix	Nitro Gel	250 mg per gel
Amix	Nitro Beet Root MaX	50 mg per capsule
HSN	Evopump powder	600 mg per 20 g
HSN	Evopump	48 mg per capsule

## Data Availability

The data presented in this study are available in the tables of this article. The data presented in this study are available on request from the corresponding author.

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
