# Peer review of "Nitric Oxide in the Field: Prevalence and Use of Nitrates by Dietitians and Nutritionists in Spanish Elite Soccer"

_nutrients, 2023, doi:10.3390/nu15245128_

Round 1

Reviewer 1 Report

Comments and Suggestions for Authors

Comments on the Quality of English Language

Minor grammatical errors to be corrected.

Author Response

Summary: The aim of this manuscript by Sebastiá-Rico et al. was to evaluate the use and behaviour’s related to nitrate consumption in Spanish elite soccer clubs. The main contribution of this manuscript is that it adds to the literature by investigating the use and behaviour’s related to nitrate consumption in Spanish elite soccer clubs, of which scientific knowledge on the use and behaviour’s of nitrate supplementation in elite soccer is currently lacking. The results of the survey data gathered from dietitians or nutritionists of elite Spanish soccer teams showed that main findings of this study were: (1) the use of nitrates as an ergogenic aid is frequent in competitions in the main Spanish professional soccer leagues but not so common in training; (2) most clubs did not take into account mouthwash control, nor did they monitor nitrates through diet; (3) nitrate loading strategies are not commonly used; (4) various forms of administration, times of consumption and doses were used. Strengths of this study include that it adds to the limited scientific literature on the use and behaviour’s related to nitrate consumption in elite level soccer players. Additionally, based on current scientific knowledge on the use of nitrates for improving athletic performance and based on the findings from the study, the study authors provide recommendations using a nice figure for the use of nitrates for improving soccer performance. The conclusions and recommendations based on the findings and recommendations for nitrate use in sports are also appropriate.

Response of the authors: We appreciate the reviewer's comments on our work, we are glad that it is of interest to you and that you found the results and the figure showing the compiled recommendations interesting.

General comments: Overall, this study was generally well conducted and written. This manuscript adds to the literature by investigating the use and behaviour’s related to nitrate consumption in Spanish elite soccer clubs, of which scientific knowledge on the use and behaviour’s of nitrate supplementation in elite soccer is currently lacking. The findings from this study provide information that although there are current science-based recommendations for the use nitrates for improving sports performance, elite level soccer athletes or dietitians/nutritionists working with elite level soccer players may not follow these recommendations for a variety of reasons or knowledge of these scientific recommendations may be lacking. The findings from this study highlight the real-world use and behaviour’s related to nitrate consumption by elite level Spanish soccer players or dieititans/nutritionists of elite level Spanish soccer teams and also highlight the need for greater education of dietitians/nutriionists of elite level Spanish soccer teams and/or elite level Spanish soccer players on the use of nitrates for improving performance. I do not have any major concerns regarding this manuscript.

Response of the authors: We appreciate the reviewer's comments and that he was interested in the work done.

Line numbers were not included so I will provide specific comments by page number and line numbers I count from the first line of each page. Page 3, line 3: The word “the” before “obtaining” can be deleted as this is a grammatical error.

Response of the authors: In accordance with the reviewer's suggestion, the word has been deleted.

Page 3, lines 35-36: The start of the sentence reads “There are clubs”, plural, but only N = 1 club reports on this. Please modify wording appropriately to be singular rather than plural.

Response of the authors: In accordance with the reviewer's suggestion, the sentence has been modified.

Figures: Please label the x-axes for all figures with titles for the benefit of the readers. The information is in the captions underneath the figures, but it would be beneficial for the readers to label the x axes with titles also.

Response of the authors: We appreciate the reviewer's comments. However, we attempted to include information on the x-axis, but found that it degraded the resolution of the graph and made it look very crowded in figures with multiple graphs. We recommend retaining the original form of the figures unless adding the information is absolutely necessary.

Section 4.3 in Discussion: The titles for sections 4.2 and 4.3 in the discussion are the same (“Adverse effects of nitrate consumption”) and I am assuming this is a grammatical error regarding section 4.3 because this text is about reasons for using nitrates for soccer performance and benefits of nitrates for soccer performance. Please modify accordingly.

Response of the authors: In accordance with the reviewer's suggestion, the title has been modified.

Conclusions, line 2: Using a word such a “variable” would be more appropriate than “versatile” given the findings, context of this sentence, and recommendations for the use nitrates for improving sports performance.

Response of the authors: Thank you for the comment, the word has been modified.

Reviewer 2 Report

Comments and Suggestions for Authors

1.       Title: it seems that the paper is about use of nitrates among nutrition professionals; please alter; dietitans and nutritionists: why 2 titles? Consider the use of just sport nutritionist

2.       Abstract: Representatives from 45 teams: players or nutrition professionals? Please indicate

3.       Introduction: instead of ‘’use’’, throughout manuscript alter to recommendation, dietary prescription, or other term, to make it clear that the authors refer to use by athletes, recommended by sport nutritionists

4.       Supplementary material: replace with the questionnaire used translated to English, also the link for google form, indicated in the Survey design

5.       Discuss the issue of highly probable co-supplementation parallel with nitrate supplementation, see for example https://www.mdpi.com/2072-6643/15/22/4838 , also the issue of adverse effects possible with overdose with nitrate salts, and provide a table showing advantages and disadvantages of recommending food vs. salts as a nitrate source

6.       Provide a table with mg of nitrates per 100 g or 100 mL of both commercial products, wholesome foods, and powders, captured within subject’s answers

Author Response

  1. Title: it seems that the paper is about use of nitrates among nutrition professionals; please alter; dietitans and nutritionists: why 2 titles? Consider the use of just sport nutritionist

Response of the authors: In accordance with the reviewer's suggestion, the title has been modified. However, we have kept the title Dietitian-Nutritionist, because in Spain one can practice as a dietitian (with a 2-year higher degree) and/or as a nutritionist (with a 4-year university degree). Given that 5 dietitians and 40 nutritionists have responded to the survey, it seems appropriate to include the title of dietitian-nutritionist, because if we only included nutritionist or sports nutritionist, we would be, technically speaking, excluding dietitians. In addition, I attach a link to the official regulations of the article that lists the competences and qualifications of dietician-nutritionists in Spain: https://www.boe.es/eli/es/o/2009/03/18/cin730

  1. Abstract: Representatives from 45 teams: players or nutrition professionals? Please indicate

Response of the authors: In accordance with the reviewer's suggestion, the sentence has been modified.

  1. Introduction: instead of ‘’use’’, throughout manuscript alter to recommendation, dietary prescription, or other term, to make it clear that the authors refer to use by athletes, recommended by sport nutritionists

Response of the authors: In accordance with the reviewer's suggestion, the sentences has been modified.

  1. Supplementary material: replace with the questionnaire used translated to English, also the link for google form, indicated in the Survey design

Response of the authors: In accordance with the reviewer's suggestion, the questionnaire has been translated and the link to the questionnaire has been changed.

  1. Discuss the issue of highly probable co-supplementation parallel with nitrate supplementation, see for example https://www.mdpi.com/2072-6643/15/22/4838 , also the issue of adverse effects possible with overdose with nitrate salts, and provide a table showing advantages and disadvantages of recommending food vs. salts as a nitrate source

Response of the authors: Thank you for the comment, information was added to the discussion regarding the combination of supplements with nitrates.

  1. Provide a table with mg of nitrates per 100 g or 100 mL of both commercial products, wholesome foods, and powders, captured within subject’s answers

Response of the authors: In accordance with the reviewer´s suggestion, a table has been added compiling some of the products featured by the brands mentioned by the respondents.